# DEEP NEURAL NETWORKS WITHOUT NORMALIZATION

## ABSTRACT

Normalization layers are ubiquitous in modern neural networks and have long been considered essential. In this work, we demonstrate that we can achieve strong performance without them, using a remarkably simple technique. We introduce Dynamic Tanh (DyT), an element-wise operation: $\mathrm{DyT}(\boldsymbol{x}) = \tanh(\alpha\boldsymbol{x})$, as a drop-in replacement to normalization layers (e.g., layer normalization). DyT is directly inspired by the simple observation that normalization layers produce tanh-like, S-shaped curves for their input-output mappings. With DyT, networks without normalization layers could match or exceed the performance of their normalization counterparts, while keeping all other training hyperparameters intact. Experiments across diverse settings validate this, ranging from recognition to generation, ConvNets to LLMs, and supervised to self-supervised learning. Our findings challenge the conventional understanding that normalization layers are indispensable, and provide new insights into their workings.

## 1 INTRODUCTION

Over the past decade, normalization layers have solified their positions as one of the most fundamental components of modern neural networks. It all traces back to Batch Normalization (Ioffe & Szegedy, 2015), which enabled drastically faster and better convergence on visual recognition models, and then quickly gained momentum. Since then, many variants for different network architectures or domains have been proposed (Ba et al., 2016; Ulyanov et al., 2016; Wu & He, 2018; Zhang & Sennrich, 2019). Today, virtually all modern networks use normalization layers, with Layer Normalization (LN) (Ba et al., 2016) being one of the most popular, particularly in Transformers (Vaswani et al., 2017).

The widespread adoption of normalization layers is largely driven by their empirical benefits in optimization (Santurkar et al., 2018; Bjorck et al., 2018). In addition to achieving lower final loss, they help accelerate and stabilize convergence. As neural networks become wider and deeper, this necessity becomes ever more critical (Brock et al., 2021a; Brody et al., 2023). Consequently, normalization layers are widely regarded as crucial, if not indispensable, for the effective training of deep neural networks. This belief is subtly evidenced by the fact that, in recent years, novel architectures often seek to replace self-attention or convolution layers, but mostly keep the normalization layers in place.

In this paper, we challenge this belief by introducing a simple alternative to normalization for deep networks. Our approach begins with the observation that layer normalization layers map their inputs to outputs with tanh-like, S-shaped curves, dynamically scaling them and then squashing the extreme values. Inspired by this insight, we propose an element-wise operation termed Dynamic Tanh (DyT), defined as: $\mathrm{DyT}(\boldsymbol{x}) = \tanh(\alpha\boldsymbol{x})$, where $\alpha$ is a learnable parameter. This operation aims to emulate the behavior of layer normalization by learning an appropriate scaling factor through $\alpha$ and squashing extreme values via the bounded tanh function. Notably, unlike normalization layers, it achieves both effects without the need to compute activation statistics.

By replacing normalization layers with DyT in architectures such as language and vision Transformers (Vaswani et al., 2017; Dosovitskiy et al., 2020), our empirical studies demonstrate that DyT can maintain training stability and achieve high final performance, across a wide range of settings. Employing DyT is straightforward for any existing architectures, and does not require additional hyperparameter tuning for training. DyT challenges the notion that normalization layers are indispensable for deep neural networks, and provides new insights into the properties of normalization layers, complementing existing theoretical understanding on normalization.

## 2 METHOD

### 2.1 WHAT DO NORMALIZATION LAYERS DO?

We first empirically study the behaviors of normalization layers in trained networks. For this analysis, we take a trained Vision Transformer model (ViT-B) (Dosovitskiy et al., 2020) on ImageNet-1K (Deng et al., 2009), and a trained wav2vec 2.0 Large model (Baevski et al., 2020) on LibriSpeech (Panayotov et al., 2015). Both models use Layer Normalization (LN).

For both trained networks, we sample a mini-batch of input data and do a standard forward pass through the network. We then measure the input and output for the norm layers, i.e., tensors immediately before and after the normalization operation, excluding the learnable scaling and shifting transformations inside these layers. Since normalization preserves the dimensions of the input tensor, we can establish a one-to-one correspondence between the input and output tensor elements, allowing for a direct visualization of their relationship.

For both models, in earlier norm layers (the first 30%-40% layers), we find this input-output relationship to be mostly linear, resembling a straight line in an $x$-$y$ plot. For deeper layers where we make more intriguing observations, the plots for four layers are shown in Figure 1 below.

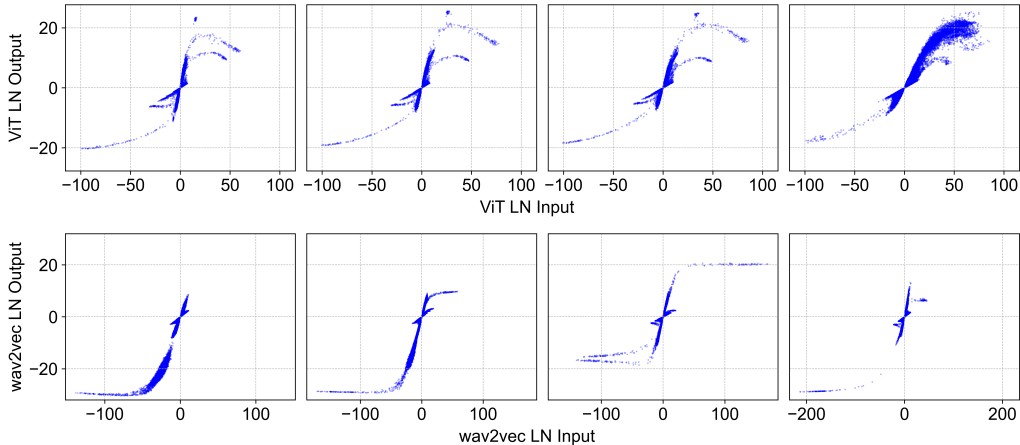

Figure 1: **Output vs. input of selected layer normalization (LN) layers in ViT and wav2vec 2.0 models.** We sample a mini-batch of data points, and plot input / output values of four LN layers in each model. The outputs are before the scaling and shifting transforms in LN. The S-shaped curves highly resemble that of a tanh function. This motivates us to propose Dynamic Tanh (DyT) as a replacement, with a learnable coefficient $\alpha$ to account for different scales on the $x$ axis.

A striking first observation is that these curves' shapes highly resemble full or partial S-shaped curves represented by a tanh function. One might expect LN layers linearly transforms the input tensor, as subtracting means and dividing by stds are linear operations. In fact, LN normalizes in a per-token manner, only linearly transforming each token's activations. As tokens have different mean and variance values, the linearity does not hold collectively on all activations of the input tensor. Nonetheless, at first sight, it is still surprising to us that the actual non-linear transformation is highly similar to a scaled tanh-function.

For such an S-shaped curve, we note that the central part, represented by points with $x$ values close to zero, is still mostly in a linear shape. Most points (∼99%) fall in this linear range. However, there are still many points that clearly fall out of this range, which are considered to have "extreme" values, e.g. those with $x$ larger than 100 or smaller than -100. For these values, norm layers' main effect is to *squash* them into less extreme values, more in line with the majority of points. This is the part where norm layers could not approximated by a simple affine transformation layer. We hypothesize this squashing effect on extreme values is what makes norm layers important and indispensable.

How does an LN layer performs a linear transformation for each token, but also squashes the extreme values in such a non-linear fashion? To understand this, we visualize the points grouped by tokens and channels respectively. This is plotted in Figure 2, by taking the third subplot for ViT from Figure 1,

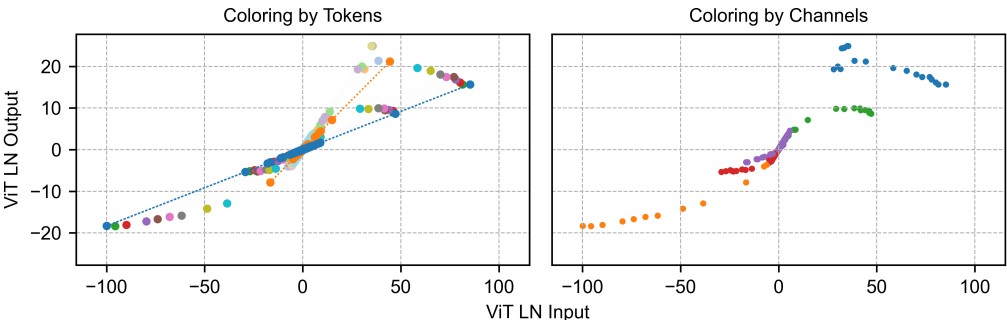

Figure 2: **Output vs. input of an LN layer, with tensor elements colored by different channel and token dimensions.** An input tensor has the shape (samples, channels, tokens), and we visualize its elements by coloring the same tokens (left) and channels (right) as the same colors. *Left*: for the same token (same color), the points from different channels form a straight line (there are dotted lines as examples), as normalization per token is a linear operation across channels. Interestingly, when plotted collectively they form a non-linear tanh-shaped curve. *Right*: each channel has input at different ranges of $x$ axis, forming a part of the collective tanh-shaped curve. Certain channels (orange, blue) tend to have more extreme $x$ values that are squashed by LN.

but with a sampled subset of points for more clarity. When we select the channels to plot, we make sure to include the channels with extreme values.

In the left of Figure 2, we visualize each token's activations using one color. We observe that all points from any single token does form a straight line. However, since each token has a different mean and variance, the slopes are different. Tokens with smaller input $x$ ranges tend to have smaller variance, and the norm layer will divide their activations using a smaller std, and hence produces a larger slope in the straight line. Collectively, they form an S-shaped curve that resembles a tanh function. In the right plot, we color each channel's activations using the same color. We find that different channel tend to have drastically different input ranges, with only a few channels (e.g., blue, orange) exhibiting large extreme values. These are the channels that get transformed the most by the norm layer, from its squashing effect.

## 2.2 DYNAMIC TANH (DyT) LAYERS

Inspired by the similarity between the shapes of normalization layers and a scaled tanh function, we propose Dynamic Tanh (DyT) as an alternative to norm layers. Given an input tensor $\boldsymbol{x}$, a DyT layer is defined as follows:

$$\mathrm{DyT}(\boldsymbol{x}) = \boldsymbol{\gamma} * \tanh(\alpha\boldsymbol{x}) + \boldsymbol{\beta} \tag{1}$$

$\alpha$ is a learnable scalar parameter that allows scaling the input dynamically based on its range, accounting for varying $x$ scales in Figure 1. $\boldsymbol{\gamma}$ and $\boldsymbol{\beta}$ are learnable, per-channel vector parameters, the same as those used in all normalization layers—they allow the output to scale back to any scales. They sometimes could be considered a separate affine layer; for our purposes, we consider them to be part of the DyT layer, just like how normalization layers also include them.

DyT is *not* a new type of normalization layer, as it operates on each input element from a tensor independently during a forward pass, without computing statistics or other types of aggregations. It does, however, preserve the effect of norm layers in squashing the extreme values in a non-linear fashion, while almost linearly transforming the very central parts of the input.

Integrating DyT layers into an existing architecture is straightforward: one DyT layer replaces every normalization layer (e.g., LN). Though DyT may look like or be considered an activation function, this study only uses it to replace normalization layers, without altering any parts of the activation functions in the original architectures, like GELU or ReLU. All other parts of networks also remain intact. We also observe there is no need to tune the training hyperparameters designed for the original architectures, for DyT to perform well.

We find initializing $\alpha$s to 1 to be sufficient in almost all cases, except training large LLMs. We always simply initialize $\boldsymbol{\gamma}$ to an all-one vector, and $\boldsymbol{\beta}$ to an all-zero vector following normalization layers.

However, when training very wide models in a under-training regime (e.g., in training LLMs), a smaller initial value for $\alpha$ (e.g., 0.2) could be more helpful. In an over-training regime, where models are trained for many epochs, initializing $\alpha$ differently from 1 only affects convergence speed without much impact on final performance. We provide detailed analysis on $\alpha$ initialization in Section 4.2.

## 3 EXPERIMENTS

We conduct experiments across four different modalities: image, language, audio, and DNA sequences, to demonstrate the effectiveness of DyT-based normalization-free networks. In each experiment, we replace the normalization layers in the original architectures with DyT layers, and then train and evaluate both versions of the models. One of our objectives is to showcase that DyT-based models could obtain comparable performance without significant changes to the training recipe and hyperparameters. Therefore, in all experiments, we use the same hyperparameters that were used for the normalized models. The only exception is the language models, where we add a learnable scalar parameter after the word embedding layer and adjust the initial value of $\alpha$ in all DyT layers. However, we still keep all other hyperparameters the same. For instructions on reproducing our experiments, please refer to Appendix A.

**Supervised image classification.** We first evaluate the performance of DyT with a standard image classification task. We train three different types of models: Vision Transformer (ViT) (Dosovitskiy et al., 2020), ConvNeXt (Liu et al., 2022), and MLP-Mixer (Tolstikhin et al., 2021), in various sizes using the ImageNet-1K dataset (Deng et al., 2009). These models were chosen for their popularity and distinct operations: attention (ViT), convolution (ConvNeXt), and pure MLP operations (MLP-Mixer). Additionally, they apply normalization layers in different locations: ViT and MLP-Mixer place layer normalization at the beginning of each residual block, while ConvNeXt places layer normalization between the convolution layers. The evaluation results are presented in table 1.

Table 1: **Supervised image classification accuracy with ImageNet-1K.** The DyT models use identical hyperparameters as their LN counterparts. DyT achieves comparable or better performance than LN across all model architectures and sizes.

| Model | LN | DyT | $\Delta$ |
|---|---|---|---|
| ViT-Base | 82.3% | 82.6% | +0.3% |
| ViT-Large | 82.6% | 82.8% | +0.2% |
| ConvNeXt-Base | 83.8% | 83.9% | +0.1% |
| ConvNeXt-Large | 84.3% | 84.4% | +0.1% |
| MLP-Mixer-Base | 78.6% | 78.4% | -0.2% |

The results demonstrate that the performance is consistently comparable between LN and DyT. This suggests that DyT can effectively replace normalization layers, regardless of the primary operations and the locations where normalization layers are applied.

**Self-supervised visual representation learning.** We next evaluate the performance of DyT in self-supervised learning paradigms. We use two self-supervised visual representation learning methods: MAE (He et al., 2022), an autoencoder method, and DINO (Caron et al., 2021), a joint embedding method. These two methods are chosen due to their own challenges. MAE includes both an encoder and a decoder with different dimensionality. It presents significant challenges for joint training both without normalization. For joint embedding methods like DINO, the encoder-only architecture often faces stability issues during training, and normalization usually helps stabilize it. Thus, evaluating DyT with these methods is crucial to demonstrating the effectiveness of DyT.

We use standard ImageNet-1K evaluation methods from both papers. The networks are first pretrained on the ImageNet-1K dataset (Deng et al., 2009). The performance of the pretrained encoders is then evaluated by attaching a classification layer, either through fine-tuning (updating both encoder and classification layer weights via gradient descent) or linear probing (freezing the encoder weights and updating only the classification layer). The results are summarized in Table 2 .

Table 2: **Self-supervised visual representation learning results with ImageNet-1K.** All models are pretrained on the ImageNet-1K training set without using any labels. The pretrained encoders are then evaluated through either fine-tuning or linear-probing. LN and DyT experiments use identical hyperparameters for each model. The table shows that DyT achieves comparable performance to LN.

| Model | LN | DyT | $\Delta$ |
|---|---|---|---|
| MAE ViT-Base (fine-tuning) | 83.6% | 83.6% | 0.0% |
| MAE ViT-Large (fine-tuning) | 85.9% | 85.9% | 0.0% |
| DINO ViT-Base/16 (linear-probing) | 78.2% | 78.1% | -0.1% |
| DINO ViT-Base/8 (linear-probing) | 80.1% | 80.1% | 0.0% |

The results demonstrate that the performance of DyT is consistently comparable to LN in self-supervised learning tasks. This suggests that the effectiveness of DyT is not influenced by the change of the learning paradigms.

**Diffusion models.** We further evaluate the effectiveness of DyT layer on vision tasks using diffusion models. Two different sizes DiT models (Peebles & Xie, 2023) are pretrained with ImageNet-1K (Deng et al., 2009). Notably, DiT uses a unique training recipe compared to other models evaluated in this paper. It uses a constant learning rate throughout the training and no weight decay. This setup tests the capability of DyT without common practices such as learning rate warmup and decay. For evaluation, the final Fréchet Inception Distance (FID) scores, computed on 50,000 images with 250 DDPM sampling steps, are reported in Table 3.

Table 3: **Diffusion model generation FID results (lower is better) with ImageNet-1K.** The LN and DyT models use identical training hyperparameters. DyT achieves improved performance with LN for diffusion models with different sizes.

| Model | LN | DyT | $\Delta$ |
|---|---|---|---|
| DiT-B/4 (FID) | 68.7 | 68.4 | -0.3 |
| DiT-L/2 (FID) | 18.2 | 18.0 | -0.2 |

The results indicate that the performance of DyT is comparable to LN. This suggests that DyT is effective for diffusion models and does not require learning rate warmup and decay, provided that its LN counterparts do not need these either.

**Language modeling.** To evaluate the effectiveness of DyT in language modalities, we test it on language modeling tasks. Specifically, two LLaMA (Touvron et al., 2023a;b) models are trained to compare the performance of DyT with normalization layers. Unlike the original Transformer (Vaswani et al., 2017), LLaMA uses a non-standard normalization layer—root mean square layer normalization (RMSNorm) (Zhang & Sennrich, 2019), along with other architectural improvements (Chowdhery et al., 2023). RMSNorm differs from LN in that it does not perform mean centering. Pretraining is conducted on the Pile (Gao et al., 2020) dataset with 300B tokens for the 1.4B model and 500B tokens for the 7B model, following the recipe from (Brown et al., 2020). In addition to measuring pre-training loss, evaluation is performed on 15 zero-shot tasks using `lm-harness` (Gao et al., 2023). Table 4 shows the comparison. The results suggest that DyT can perform comparably to normalization layers like RMSNorm for language modeling. As we stated at the beginning of the section, we have to make some changes to the network and adjust the initialization value of $\alpha$, Please refer to 4.2 for a more detailed explanation of the modification.

Table 4: **Language modeling zero-shot results with 15 `lm-harness` tasks.** All models are pretrained with 500B tokens from the Pile dataset. We report the average accuracy (higher is better) on 15 zero-shot tasks from `lm-harness`, and the pre-training loss (lower is better). DyT achieves comparable performance to RMSNorm.

| Accuracy / Loss | RMSNorm | DyT | $\Delta$ |
|---|---|---|---|
| LLaMA-1.4B | 45.1% / 2.06 | 45.0% / 2.14 | -0.1% |
| LLaMA-7B | 49.3% / 1.92 | 49.3% / 1.87 | 0.0% |

**Audio waveform pretraining.** We further evaluate the effectiveness of DyT by pretraining the wav2vec 2.0 (Baevski et al., 2020) model, a standard speech representation learning model, on the LibriSpeech (Panayotov et al., 2015) dataset. We adopted two setups of the wav2vec 2.0 architecture: pre-norm and post-norm, which place the normalization layer at the beginning or the end of the blocks, respectively. After pretraining for 200 epochs, we report the evaluation loss in Table 5. The results show that DyT performs on par with LN for audio waveform pretraining tasks.

Table 5: **Audio waveform pretraining validation loss (lower is better) on LibriSpeech.** The models are pretrained with LibriSpeech dataset, and the validation losses at epoch 200 are reported. The LN and DyT experiments use identical hyperparameters. The table shows that DyT achieves comparable performance to LN for wav2vec 2.0 models with different normalization layer positions.

| Model | LN | DyT | $\Delta$ |
|---|---|---|---|
| wav2vec 2.0 Base (Pre-Norm) | 2.14 | 2.15 | +0.01 |
| wav2vec 2.0 Base (Post-Norm) | 2.19 | 2.15 | -0.04 |

**DNA sequence pretraining.** For experiments on DNA Sequences, we pretrain HyenaDNA (Nguyen et al., 2024) model with human reference genome (GRCh38, 2013), and test the downstream task performance with GenomicBenchmarks (Grešová et al., 2023). The results is presented in Table 6. These results illustrate that DyT can maintain or slightly enhance performance compared to LN.

Table 6: **GenomicBenchmarks results with pretrained HyenaDNA model.** The HyenaDNA model is first pretrained with the human reference genome. Evaluation is performed by fine-tuning the pretrained encoder with each data from the genomic benchmarks. The LN and DyT experiments for each model use identical hyperparameters. The table shows that DyT achieves comparable performance to LN for different downstream tasks.

| Task | LN | DyT | $\Delta$ |
|---|---|---|---|
| Mouse Enhancers | 85.1% | 85.1% | 0.0% |
| Coding vs Intergenomic | 91.3% | 91.4% | +0.1% |
| Human vs Worm | 85.9% | 85.9% | 0.0% |
| Human Enhancers Cohn | 74.2% | 74.4% | +0.2% |
| Human Enhancers Ensembl | 89.2% | 89.2% | 0.0% |
| Human Regulatory | 93.8% | 93.7% | -0.1% |
| Human Non-tata Promoters | 96.6% | 96.5% | -0.1% |
| Human OCR Ensembl | 80.9% | 80.9% | 0.0% |

## 4 ANALYSIS

### 4.1 UNDERSTANDING THE ROLE OF $\alpha$

**Correlation between final $\alpha$ and $1/\text{std}$ of activation.** We conducted further analysis on the role of $\alpha$ for pretrained networks. The investigation reveals that $\alpha$ adapts to learn the inverse of the standard deviation of the input activations. Figure 3 illustrates this relationship, demonstrating that the values of $\alpha$ across different DyT layers correlate with the inverse of the standard deviation of the layer inputs for two different models. This indicates that $\alpha$ could help manage larger activations by scaling them down, effectively preventing saturation.

**Increasing activation std with depth.** Moreover, we observe that deeper layers tend to have larger standard deviations in their input activations. Such an increasing standard deviation with depth is potentially an important feature of deep residual networks, as pointed out by Brock et al. (2021a). This could also explain why the static hyperbolic tangent function does not perform as well as DyT, as it cannot adapt to the changing activation distributions across layers.

**Dynamic adaptation of $\alpha$ during training.** We also observe that the learned value of $\alpha$ closely tracks the standard deviation of activations throughout training. As shown in Figure 4, the inverse of $\alpha$ fluctuates in response to changes in activation standard deviation, further supporting the dynamic role of $\alpha$ in maintaining stable and effective training.

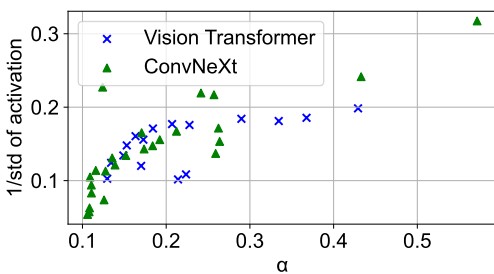 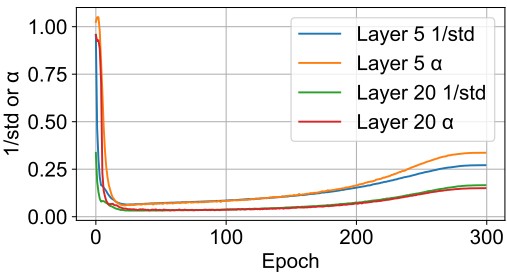

Figure 3: **The final $\alpha$ values and the $1/\text{std}$ of the activation are correlated.** We plot the $\alpha$ values of two pretrained models, ViT and ConvNeXt, with the inverse of the standard deviation of the input activation. The graph shows that the learned $\alpha$ are mostly correlated with inverse of standard deviation of the activation.

Figure 4: **The learned $\alpha$ and the $1/\text{std}$ of input activation during training** We pick two DyT layers from the ViT-Base model and record the inverse of standard deviation of the input and the learned $\alpha$ at end of each epoch. It shows that $\alpha$ values and the standard deviation of the activation change together during training.

### 4.2 INITIALIZATION OF $\alpha$

For the initialization of the learnable scalar $\alpha$, we find that setting it to $1$ works well in most cases. While adjusting the initial value of $\alpha$ can lead to faster early convergence, this advantage typically does not carry over to the later stages of training. However, in LLMs, the proper initialization of $\alpha$ proves to be important, as early improvements tend to influence the final performance. We suspect this difference arises because language modeling often operates in an underfitting regime, unlike other tasks where overfitting is a dominant issue.

**Learnable scaling after input embeddings.** In our implementation of LLaMA (Touvron et al., 2023a;b) models with DyT, we introduce a learnable scaling scalar immediately after the word embedding layers, initialized to $\sqrt{d_{\text{width}}}$, where $d_{\text{width}}$ represents the model's hidden dimension. Without this scaling scalar, training struggled to progress meaningfully in the early stages. The underlying issue could be traced to the small magnitude of activations at the start of training (around 0.02), and it is mainly caused by the small magnitude outputs of the word embedding layers at initialization. By adding a learnable scalar, we mitigated the problem, allowing the model to converge more quickly. This approach is similar to the original Transformer architecture (Vaswani et al., 2017), which uses a fixed scaling parameter $\sqrt{d_{\text{width}}}$ at the start.

Notably, this issue primarily exists in models with embeddings as inputs. In contrast, models that start with linear or convolutional layers typically produce outputs from the first layers with significantly larger magnitudes than the initialization value of , without the scaling issue. We verified this using a ViT model with discrete token embedding as the input as well.

**Optimal initial value of $\alpha$ for LLMs.** After adding the scaling for the word embedding layers, we conduct a series ablation studies on the optimal values of $\alpha$ for a different configurations of the LLaMA models. In all the ablation studies, we train the networks for 10,000 steps and compare the losses at that point. For each configuration we experiment with 6 different possible initial values of $\alpha$: 2.0, 1.0, 0.5, 0.2, 0.1, 0.05.

Table 7: **Optimal initial value of $\alpha$ vs. the depth and width of the LLaMA model.** We train each model configuration with 6 different initial values of $\alpha$ : 2.0, 1.0, 0.5, 0.2, 0.1, 0.05. Each training ran $10,000$ steps, and we report the initial value that produce the lowest loss.

| Width / Depth | 8 | 16 | 24 | 32 | 40 |
|---|---|---|---|---|---|
| 1024 | 1.0 | 1.0 | 1.0 | 1.0 | 1.0 |
| 2048 | 0.5 | 0.5 | 0.5 | 0.5 | 0.5 |
| 3072 | 0.2 | 0.2 | 0.2 | 0.2 | 0.2 |
| 4096 | 0.2 | 0.1 | 0.1 | 0.1 | 0.2 |
| 5120 | 0.1 | 0.1 | 0.1 | 0.1 | 0.1 |

Table 7 presents the results showing the influences of model depth and width on the optimal initial value of $\alpha$. It shows a clear trend: the depth of the models does not influence the choice of the optimal $\alpha$, while the width of the models has a significant impact on the optimal initial value of $\alpha$.

After establishing that the width of the network is the dominant factor in choosing the optimal $\alpha$ initialization value, we conducted two further ablation studies on the attention head dimension and the length of input sequences. We discover that the head dimension and the length of the input sequences have no clear evidence of influencing the choice of the optimal $\alpha$ initialization value. We list the results in the Appendix B.1 for completeness.

To obtain more practical guidance on the optimal $\alpha$ initialization value, we carefully searched for the optimal $\alpha$ using a shallow model (8 layers) with model widths ranging from 512 to 8192. We plot the optimal values in Figure 5, which shows a clear trend that, the wider the network, the smaller the initialization value of $\alpha$ should be.

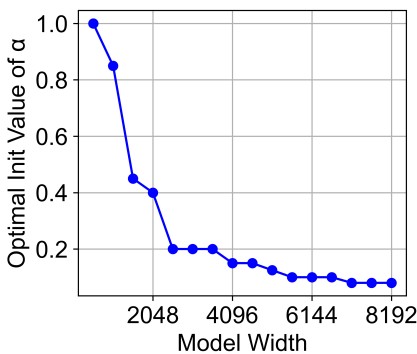

Figure 5: **Optimal initial value of $\alpha$ vs. model width.** As the model becomes wider, the optimal initial value of $\alpha$ decreases.

### 4.3 THE IMPORTANCE OF SQUASHING AND $\alpha$

To further understand the importance of the squashing effect and the learnable parameter $\alpha$ in DyT, we conduct a number of experiments to assess the model's performance without $\alpha$ and without functions that provide a squashing effect. We used a standard ViT-Tiny model and replaced its normalization layers with four different functions: identity, tanh, hardtanh, and sigmoid. For each function, we conducted two sets of experiments: one with the learnable parameter $\alpha$ and one without it. The result is listed in Table 8.

Table 8: **ViT-Tiny image classification results on ImageNet-1K** We replace the layer normalization layers with each function listed in the table. The results show that both the squashing effect and the learnable parameter $\alpha$ are essential for training effective models.

| Model / Function | identity | tanh | hardtanh | sigmoid |
| --- | --- | --- | --- | --- |
| without $\alpha$ | Diverge | 69.2% | 68.7% | 66.3% |
| with $\alpha$ | Diverge | 73.5% | 71.7% | 70.2% |

The results indicate that the squashing effect is a key factor in stabilizing training. When using the identity function, the model's training was unstable and diverged. In contrast, functions that provide a squashing effect, such as tanh, hardtanh, and sigmoid, enabled stable training without divergence.

Moreover, the choice of squashing function significantly impacts performance. Sigmoid, for example, yielded the lowest accuracy, likely due to its tendency to center mean activations around 0.5 rather than 0. Similarly, hardtanh performed worse than tanh, suggesting that the optimal squashing effect lies within a specific range. These findings underscore the critical role of the squashing effect in stabilizing training, and highlight the importance of learnable parameter $\alpha$ to control this effect.

## 5 RELATED WORK

**Normalization Layers.** Normalization techniques are fundamental in deep learning, starting with Local Response Normalization (Lyu & Simoncelli, 2008; Jarrett et al., 2009) in models like AlexNet (Krizhevsky et al., 2012). Batch normalization (Ioffe & Szegedy, 2015) popularized normalization by enhancing convergence and generalization through mini-batch activation normalization. It leads to various methods targeting different data dimensions—channel (Ba et al., 2016; Zhang & Sennrich, 2019), spatial/temporal (Ulyanov et al., 2016), or both (Ba et al., 2016; Wu & He, 2018). In transformer models (Vaswani et al., 2017; Dosovitskiy et al., 2020), layer normalization(Ba et al.,

2016) has become the primary normalization strategy. Recently, rms normalization (Zhang & Sennrich, 2019), used in models like T5 (Raffel et al., 2020) and LLaMA (Touvron et al., 2023a), enhances layer normalization by omitting mean centering, highlighting the ongoing evolution of normalization techniques in deep learning.

**Benefits of Normalization.** Early research on benefits of normalization predominantly centered on Batch Norm, elucidating its capacity to enhance model training and performance through various mechanisms. These advantages include propagating informative activation patterns into deeper layers, which maintains gradient flow during training (Daneshmand et al., 2020; Balduzzi et al., 2017). Normalization also reduces dependency on initialization schemes, making networks less sensitive to initial weights (De & Smith, 2020; Shao et al., 2020; Zhang et al., 2019). It accelerates convergence by moderating outlier eigenvalues that can impede learning (Karakida et al., 2019; Bjorck et al., 2018). Additionally, normalization effectively auto-tunes learning rates, similar to adaptive optimizers (Arora et al., 2018; Tanaka & Kunin, 2021), and smooths the loss landscape for more stable optimization (Santurkar et al., 2018; Yong et al., 2020). These properties collectively enhance training robustness and efficiency across architectures and applications.

With transformer models' advent (Vaswani et al., 2017), research shifted focus to LayerNorm (Ba et al., 2016). LayerNorm operates across features of a single sample, unlike Batch Norm's batch dimension, making it well-suited for sequential data and enhancing transformer performance in natural language tasks (Xiong et al., 2020; Nguyen & Salazar, 2019). LayerNorm stabilizes transformer training by mitigating internal covariate shift, facilitating faster convergence and improved generalization (Xu et al., 2019). It also alleviates vanishing and exploding gradients in deep networks (Nguyen & Salazar, 2019). Furthermore, LayerNorm's per-sample normalization statistics enable effective learning of complex distributions, making it valuable for modeling long-range dependencies (Xiong et al., 2020).

**Normalization-free networks.** The research on Normalization-free networks challenges the belief that normalization layers are indispensable for the effective training of deep neural networks. This domain seeks to match the performance of traditional models while using normalization, thereby streamlining architectures and addressing issues inherent to normalization layers (Brock et al., 2021a).

A pioneering study by Brock et al. (Brock et al., 2021a;b) highlighted the potential of training high-performance ResNet models without normalization (Smith et al., 2023). They introduced a meticulously crafted initialization scheme (De & Smith, 2020), coupled with weight normalization techniques (Huang et al., 2017; Qiao et al., 2019), and a novel training methodology that incorporates very strong data augmentation (Cubuk et al., 2020), intensive regularization (Srivastava et al., 2014; Huang et al., 2016), and adaptive gradient clipping (Brock et al., 2021b). This approach not only achieved high accuracy but also demonstrated superior generalization on out-of-distribution data.

Another line of research focuses on modifying transformer blocks to reduce dependency on normalization and skip connection (He et al., 2023; He & Hofmann, 2023). These studies explore the feasibility of omitting normalization from certain parts of transformer blocks, although they acknowledge the necessity of retaining layer normalization in either the encoder or decoder to maintain functional models. Other research has been exploring alternative strategies, such as novel initialization methods, to facilitate normalization-free training. Approaches like FixUp (Zhang et al., 2019), ReZero (Xiong et al., 2020), and SkipInit (De & Smith, 2020) focus on adjusting weight initialization to support training without normalization. However, these methods were not shown to work across various modern networks, most notably large Transformers.

## 6 CONCLUSION

In this work, we introduced Dynamic Tanh (DyT), a simple alternative to traditional normalization layers in deep neural networks. DyT dynamically adjusts the input activations via a learnable scaling factor $\alpha$ and squashing the extreme values through a tanh function, effectively capturing the behavior of normalization while simplifying the architecture. Through experiments across a wide range of modalities, including image, audio, language, and genomics, our results demonstrate that DyT not only matches the performance of traditional normalization techniques but also ensures training stability without the need for extensive hyperparameter tuning.

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

# A  IMPLEMENTATION DETAILS

## A.1  SUPERVISED IMAGE CLASSIFICATION

For all supervised image classification experiments, we employed a standardized recipe, detailed in Table 9, for each model listed. This recipe is primarily adapted from the one used by ConvNeXt (Liu et al., 2022), as it demonstrates superior performance compared to the original recipes utilized in DeiT (Touvron et al., 2021) and MLP-Mixer (Tolstikhin et al., 2021).

Table 9: Supervised Image Classification Training Recipe with ImageNet-1K

|  | ViT-B | ConvNeXt-B | ConvNeXt-L | Mixer-B |
|---|---|---|---|---|
| Epochs | 300 | 300 | 300 | 300 |
| Warmup Epochs | 20 | 20 | 20 | 20 |
| Optimizer | AdamW | AdamW | AdamW | AdamW |
| Batch Size | 4096 | 4096 | 4096 | 4096 |
| LR | $4.10^{-3}$ | $4.10^{-3}$ | $4.10^{-3}$ | $4.10^{-3}$ |
| LR Decay | cosine | cosine | cosine | cosine |
| Weight Decay | 0.05 | 0.05 | 0.05 | 0.05 |
| Betas | (0.9, 0.999) | (0.9, 0.999) | (0.9, 0.999) | (0.9, 0.999) |
| Global Pool | ✓ | ✓ | ✓ | ✓ |
| LayerScale | ✗ | ✓ | ✓ | ✗ |
| Label Smoothing | 0.1 | 0.1 | 0.1 | 0.1 |
| Stoch. Depth | 0.1 | 0.5 | 0.5 | 0.1 |
| Gradient Clip. | ✗ | ✗ | ✗ | 1.0 |
| RRC | ✓ | ✓ | ✓ | ✓ |
| H. Flip | ✓ | ✓ | ✓ | ✓ |
| Rand Augment | 9/0.5 | 9/0.5 | 9/0.5 | 9/0.5 |
| Mixup Alpha | 0.8 | 0.8 | 0.8 | 0.8 |
| Cutmix Alpha | 1.0 | 1.0 | 1.0 | 1.0 |
| Erasing Prob. | 0.25 | 0.25 | 0.25 | 0.25 |
| ColorJitter | ✗ | ✗ | ✗ | ✗ |
| Test Crop Ratio | 0.875 | 0.875 | 0.875 | 0.875 |

## A.2  LANGUAGE MODELING

For language modeling, we followed the recipe from (Brown et al., 2020) when training on the Pile (Gao et al., 2020). We used the PyTorch code base FMS FSDP (Stack, 2024) and conducted experiments on GPUs. The default initial LR is $3.10^{-3}$, and weight decay 0.1. We used batch size 256, so there is about 1M tokens per step. For evaluation, we choose 15 zero-shot commonsense reasoning tasks from `lm-harness` (Gao et al., 2023), which are: `anli_r1`, `anli_r2`, `anli_r3`, `arc_challenge`, `arc_easy`, `boolq`, `hellaswag`, `openbookqa`, `piqa`, `record`, `rte`, `truthfulqa_mc1`, `truthfulqa_mc2`, `wic`, `winogrande`. The selection is closely following LLaMA (Touvron et al., 2023a) and we simply take the average across all the metrics following common practice.

## A.3  OTHER TASKS

For all other tasks, MAE (He et al., 2022), DINO (Caron et al., 2021), DiT (Peebles & Xie, 2023), Wav2Vec 2.0 (Baevski et al., 2020), and HyenaDNA (Nguyen et al., 2024). We directly use the publicly released code from the authors without performing any hyperparameter tuning, using the original hyperparameters provided. The only modification we made was replacing the normalization with an layer. Following this adjustment, we executed the models according to the authors' instructions. For completeness, we list all the hyperparameters used by the original authors for each model below.

**MAE**  For pretraining, we used a total batch size of 4096 with a base learning rate of 1.5e-4 and a weight decay of 0.05. Training was conducted over 800 epochs with 40 warmup epochs, using a mask ratio of 0.75. For fine-tuning, we used a batch size of 16 over 50 epochs with a base learning rate of 1e-3. The same setup was applied for both ViT-base and ViT-large.

**DINO**  For pretraining, we used a total batch size of 1024 with a base learning rate of 7.5e-4 and a weight decay of 0.04. Training was conducted over 400 epochs with 10 warmup epochs. For learning probing, we used a batch size of 1024 with a base learning rate of 0.001 over 100 epochs.

**DiT**  For pretraining, we used a batch size of 256 with a learning rate of 0.1 and no weight decay. Training was conducted over 1400 epochs without any warmup epochs. For evaluation, we used 250 sampling steps with an image size of 256.

**wav2vec 2.0**  We used a batch size of 64 with a learning rate of 0.001 and a weight decay of 0.01. Training was conducted over 200 epochs with 32000 warmup steps.

**HyenaDNA**  For pretraining, we used a batch size of 1024 and a sequence length of 600 with a learning rate of 1e-3 and a weight decay of 0.2. For evaluation, we used the Genomic Benchmarks (Grešová et al., 2023) with a maximum length of 500.

## B  OTHER ABLATION STUDIES

### B.1  ABLATIONS FOR OPTIMAL INITIAL VALUE OF $\alpha$

We conducted further ablations on the influences of head dimensions and sequence length to the optimal initial value of $\alpha$. Since we have already established that the model depth doesn't have noticeable effect to the choice of optimal initial value of $\alpha$, so all the following ablation is conducted with a shallow network (8 layers).

Table 10: **Optimal initial value of $\alpha$ vs. the head dimension and the sequence length.** We train each model configuration with 6 different initial values of $\alpha$: 2.0, 1.0, 0.5, 0.2, 0.1, 0.05. Each training ran $10,000$ steps, and we report the initial value that produce the lowest loss.

| Head Dim | Seq Length / Width | 512 | 1024 | 1536 | 2048 | 2560 | 3072 | 3584 | 4096 |
|---|---|---|---|---|---|---|---|---|---|
| 32 | 4096 | 1.0 | 1.0 | 0.5 | 0.5 | 0.2 | 0.2 | 0.2 | 0.1 |
| 64 | 4096 | 1.0 | 1.0 | 0.5 | 0.5 | 0.2 | 0.2 | 0.2 | 0.1 |
| 128 | 4096 | 1.0 | 1.0 | 0.5 | 0.5 | 0.2 | 0.2 | 0.2 | 0.2 |
| 128 | 1024 | 1.0 | 1.0 | 0.5 | 0.5 | 0.2 | 0.2 | 0.2 | 0.1 |
| 128 | 2048 | 1.0 | 1.0 | 0.5 | 0.5 | 0.2 | 0.2 | 0.2 | 0.1 |
| 128 | 4096 | 1.0 | 1.0 | 0.5 | 0.5 | 0.2 | 0.2 | 0.2 | 0.2 |

From table 10, we could clearly see that the head dimension and sequence length also have negligible effect on the optimal choice of initial value of $\alpha$.

### B.2  REPLACING BATCH NORMALIZATION WITH DYT

Building on our previous experiments demonstrating DyT as an effective replacement for layer normalization, we explored its applicability to batch normalization (BN) in classic CNN architectures like ResNet-50 (He et al., 2016) and VGG16 (Simonyan & Zisserman, 2014). Additionally, we examined the effects of substituting layer normalization with batch normalization and DyT in the ViT-Base model. All models were trained from scratch on the ImageNet-1K dataset under identical conditions to isolate the impact of the normalization methods.

Our results showed that replacing batch normalization with DyT in ResNet-50 led to a decrease in accuracy, while substituting batch normalization with layer normalization caused the training to diverge. In VGG16, a small performance drop occurred with DyT, and a larger drop with layer normalization. Conversely, in ViT-Base, replacing layer normalization with batch normalization

Table 11: **Image classification results with BN, LN and DyT** We replace the BN layers with LN or DyT for both ResNet-50 and VGG16 models. And we replace the LN layers with BN or DyT layers for the ViT model.

| Model | BN | LN | DyT |
|---|---|---|---|
| ResNet-50 | 76.1% | Diverge | 74.1% |
| VGG16 | 73.3% | 70.2% | 72.1% |
| ViT-Base | Diverge | 82.3% | 82.6% |

resulted in divergence. These findings suggest that DyT can partially substitute for batch normalization in certain CNNs but doesn't fully replicate its stabilization and performance benefits. The divergence highlights the critical role of batch-dependent normalization in CNNs, which isn't addressed by layer normalization or DyT. Since batch normalization computes statistics for each channel, it lacks the squashing effect characteristic of layer normalization. This indicates that despite both are normalization layers, batch normalization and layer normalization behave differently, and DyT aligns more closely with layer normalization.

