# OpenReview forum: "Deep Neural Networks without Normalization"
_ICLR.cc/2025/Conference — ICLR 2025 Conference Withdrawn Submission_

### Official Review · Reviewer_7Gib · 2024-10-15

**Soundness:** 3
**Presentation:** 3
**Contribution:** 3
**Rating:** 5
**Confidence:** 4

**Summary:**

The paper introduces Dynamic Tanh (DyT).
DyT(x) = \gamma * tanh(\alpha x) + \beta, where \alpha is a learnable parameter.
The authors note that normalization layers produce tanh-like, S-shaped curves for their input-output mappings so they replaced normalization layers with DyT in various architectures and tasks, including:
Image classification (ViT, ConvNeXt, MLP-Mixer), Self-supervised visual learning (MAE, DINO), Diffusion models (DiT), Language modeling (LLaMA), Audio waveform pretraining (wav2vec 2.0), DNA sequence pretraining (HyenaDNA)

The authors do some analysis on what \alpha learns and note that alpha's init is important.

The TLRD: DyT is a simple alternative to normalization that can be easily integrated into existing architectures without significant hyperparameter tuning.

**Strengths:**

The method is simple, (as far as I can tell) novel, and broad applicability.
The paper verifies that it is applicable in many different domains and requires minimal modification to the network.
Also the method is computationally efficient as opposed to using norm.

**Weaknesses:**

(1) The work looks at this methods efficacy for standard model sizes at standard training lengths. There is little analysis for longer training lengths and larger model sizes. The work doesn't push the method to failure (or to large enough scale that we'd see failure).

(2) The work does a good job outlining a lot of other norm free approaches in the related works sections, but does not implement and test against ANY of them...



EDIT (per feedback from Associate Program Chairs):

(edit for 1) off the top of my head, I'd like to see an LLM.int8() style analysis on the norm vs DyT network to see if DyT is more or less numerically stable.

Similarly, I'd like to see a general analysis of numerical stability of this method vs just having normalization. Anything that would help me understand how low precision training (fp8) and quantization would be effected by using DyT.



(edit for 2) the paper says: "Approaches like FixUp (Zhang et al., 2019), ReZero (Xiong et al., 2020), and SkipInit (De & Smith, 2020) focus on adjusting weight initialization to support training without normalization. However, these methods were not shown to work across various
modern networks, most notably large Transformers."

(a) ReZero does show results on Transformers
(b) FixUp and SkipInit do not show results on Transformers but why wouldn't it work. At my previous job, I had colleges who had tested these ideas and they worked
(c) personally, I'd like to see a NormFree-ResNets adaptation for Transformers (or at least an attempt) before saying that it does not work.

In general, replacing norm with a non-lin is computational advantageous to having a norm, but entirely removing ops (and instead messing with init and adding regularization) is even more advantageous.

Side note: your citation for ReZero is wrong...

**Questions:**

The work does a good job outlining a lot of other norm free approaches in the related works sections, but does not implement and test against ANY of them...
I will give this paper a score of 5, but will change the score to a 6 if you implement and run testing against other methods (not just norm); if the testing is very thorough, I'd even consider changing the score to 8.

---

### Official Review · Reviewer_hDeh · 2024-10-25

**Soundness:** 2
**Presentation:** 2
**Contribution:** 1
**Rating:** 3
**Confidence:** 4

**Summary:**

This paper proposes Dynamic Tanh (DyT) as a replacement for normalization layers, inspired by the observation that the input-output mapping curves of layer normalization (LN) layers resemble those of tanh function. The authors demonstrate that networks using DyT in place of layer normalization can achieve performance comparable to those with layer normalization, without altering any other training hyperparameters. They also show that the learnable scalar parameter $α$ in DyT correlates with the inverse of the input activation's standard deviation, suggesting its role in stabilizing training. To further support this, they compare the performance of the ViT-Tiny model with and without $α$, highlighting the difference.

**Strengths:**

- DyT can be easily applied as a simple replacement for layer normalization.
- The authors conduct experiments across various modalities, where DyT is applied as a replacement for layer normalization.

**Weaknesses:**

- The performance of models using DyT shows no significant improvement over models using layer normalization (LN). Additional evidence is needed to demonstrate why adopting DyT would be advantageous given its comparable performance.
- While the authors mention DyT's benefit of not needing to compute activation statistics, experiments illustrating this advantage are lacking. Demonstrating how this impacts computational efficiency compared to LN would strengthen the case for DyT.
- The authors also mention that DyT has the advantage of ensuring training stability; however, there are no experiments demonstrating the extent to which DyT improves training stability compared to LN. Additional experiments, such as comparing the training and validation error curves of models using DyT and LN to illustrate how much DyT contributes to faster convergence than LN, would be beneficial.

**Questions:**

Please refer to the weaknesses section.

---

### Official Review · Reviewer_KpoY · 2024-10-29

**Soundness:** 2
**Presentation:** 3
**Contribution:** 3
**Rating:** 6
**Confidence:** 3

**Summary:**

This paper observes that layer normalization transforms the inputs following an S-shaped curve, similarly to the tanh function. It thus proposes Dynamic Tanh (DyT) that uses tanh function with learnable scalar for its input. Experiments across multiple data modalities, architectures, and learning paradigms demonstrate that DyT matches the performance of layer normalized (or RMsNormed) networks (including small-scale LLM). It also introduces some interesting observations for understanding the functionality of layer normalization.

**Strengths:**

This paper is well written with clear introduction, sufficient related work. It is good to see Figure 1 and 2 that provide clear evidence for the motivation of this paper. The overall method and experimental results are new to me, and can provide new clues to help understand layer normalization (RMSNorm). It is glad to see that this paper conducts extensive experiments to provide the comparison between DyT and LN/RMSNorm, especially the experiments on small-scale LLM.

**Weaknesses:**

1.My main concern is that some claims/descriptions are not rigorous and needs to be improved.
(1)	In the abstract, this paper states that “DyT is directly inspired by the simple observation that normalization layers produce tanh-like, S-shaped curves for their input-output mappings.” I think, the observation is from the layer normalization (like specified in the introduction, Line 41), rather than from other more general normalization. It is clear Batch normalization will have this observation (Figure 1), due to its linear-only operation in transforming data.
(2)	In Line 92~98, this paper uses some description to show the `surprise’ of the nonlinearity of layer normalization. It is not surprised due to  the nonlinearity of layer normalization is well illustrated in [1], where how the projection is performed in LN and how it affects the representation capacity of a network. I think it is  better to refer/cite the exist work [1] for credits.


2.It is glad to see this paper provides the experiments for training of small-scale LMMs (1.4B~7B). However, the evaluation for 15 lm-harness tasks seems to be weak to support the effectiveness, it is better to see the evaluation on some famous LLM-bechmark, e.g., MMLU/MMLU-Pro, where the performance of various models are evaluated. Besides, while this paper reports the loss in the end, it is better to provide the figure of loss during training (e.g., put it in the appendix.).


3.I am interested in the weakness/limit of the DyT, compared to LN. Is DyT sensitive to the learning rate (or other hyper-parameters), compare to LN? Based on the analyses on experiments of LLM, I feel that DyT is not as good as to LN, in terms of the training stability.


Ref:
[1] On the nonlinearity of Layer Normalization.  ICML 2024

**Questions:**

1.In Figure 4, it seems that learnable $\alpha$ can obtain the same effect. If this is the case, why need the squash functions? I think this paper needs to well illustrate it.

---

### Official Review · Reviewer_g8d4 · 2024-10-31

**Soundness:** 2
**Presentation:** 3
**Contribution:** 2
**Rating:** 3
**Confidence:** 4

**Summary:**

This paper proposes a novel technique called Dynamic Tanh (DyT), which is introduced as a replacement for normalization layers.
DyT is inspired by the tanh-like input-output mappings of the transformer's Layer Normalisation (LN) layers.
It is defined as a scale-shift form of tanh activation of scaled inputs. The scaling parameters are learned during backpropagation.
DyT's performance is evaluated on tasks such as supervised image classification, self-supervised learning, Diffusion models, Language modelling, Audio pretraining, and DNA sequence pretraining.
The authors report performance comparable to Layer Norm (LN) in the proposed experimentation framework.
The paper discusses the initializations of input scaling factor alpha for language models.
Furthermore, it reports that the converged alphas are equivalent to the inverse of the standard deviation of corresponding inputs.
Finally, the study reports the importance of tanh squashing in identity, hard-tanh and sigmoid activations.

**Strengths:**

The paper discovered that Layer Norm (LN) produces an "S" shaped input-output mapping. The proposed Dynamic Tanh (DyT) attempts to mimic LN's behaviour at a low computational cost. Arguably DyT is an element-wise operation and, therefore, relatively cheap. Notably, DyT can simply replace LN without additional requirements. DyT's evaluation of a wide range of tasks is an added bonus to the paper. Overall, the presented experimental framework emphasise the argument of replacing LN with DyT.

**Weaknesses:**

Despite the experimental evidence, the effect of DyT on gradients and usage of large learning rates is unexplored in the paper. This could be crucial for understanding its performance and stability under various training conditions. Following prior work in [1], a study on the effect of gain and bias on the performance of DyT could provide valuable insights into the contributions of the tanh function. Along with min-max, and z-score normalization, the use of Tanh for normalisation is not new. It was already explored for feature scaling [2, 3]. Particularly, the traditional tanh normalization scheme is non-adaptive. [4] proposed the use of adaptive tanh normalization for deep learning framework. Including this line of research in the paper would strengthen the paper.

Related work:
See questions.

**Questions:**

1. Each transformer block consists of two individual LN layers. Do both the layers exhibit the "S" shaped behaviour?
2. Figure:1 demonstrate the LN behaviour for ViT-B and wav2vec 2.0 models. The comparisons in Table:1 include ConvNext model. Does the same observations hold for the ConvNext model?
3. Traditionally, each LN layer is placed before Multi-Headed Self-Attention (MHSA) and Feed-Forward Networks(FFN). In Swin-V2 [5], LN is placed after MHSA and FFN. How does this change of position in DyT affect the convergence?
4. Line-344 states that "adjusting the initial value of alpha can lead to faster early convergence". Is there an evidence for this claim?. Moreover, how does the network optimization converge well in case of a larger alpha value?
5. Following LN [6], does the proposed DyT demonstrate invariance with respect to weight and dataset rescaling? If not, how sensitive is it to these variations?
6. Line-212 "The networks are pretrained" and Line-255 "Pretraining is conducted on", what normaization was used for the pertaining?
7. In case of DiT, only LN is replaced with DyT?. Because DiT uses scale-shift norm conditioned on time and text in addition to LN. Has the scale-shift norm also been replaced?
8. Other functions such as $\frac{x}{1+|x|}$ also produce "S" shaped curve [7] , are these considered as alternatives to tanh? If not, can these functions offer similar performance?

References:
1. Xu, J., Sun, X., Zhang, Z., Zhao, G., & Lin, J. (2019). Understanding and improving layer normalization. Advances in neural information processing systems, 32.
2. Jain, A., Nandakumar, K., & Ross, A. (2005). Score normalization in multimodal biometric systems. Pattern recognition, 38(12), 2270-2285.
3. Ribaric, S., & Fratric, I. (2005, October). A matching-score normalization technique for multimodal biometric systems. In Proceedings of Third COST 275 Workshop-Biometrics on the Internet (pp. 55-58). UK: University of Hertfordshire.
4. Cuevas, Felip Guimerà, and Helmut Schmid. "Robust Non-linear Normalization of Heterogeneous Feature Distributions with Adaptive Tanh-Estimators." International Conference on Artificial Intelligence and Statistics. PMLR, 2024.
5. Liu, Z., Hu, H., Lin, Y., Yao, Z., Xie, Z., Wei, Y., ... & Guo, B. (2022). Swin transformer v2: Scaling up capacity and resolution. In Proceedings of the IEEE/CVF conference on computer vision and pattern recognition (pp. 12009-12019).
6. Lei Ba, J., Kiros, J. R., & Hinton, G. E. (2016). Layer normalization. ArXiv e-prints, arXiv-1607.
7. https://www.wolframalpha.com/input?i2d=true&i=Divide%5Bx%2C1%2B%7Cx%7C%5D

---

### Note · Authors · 2024-11-14

**Comment:**

We would like to thank reviewers for their constructive feedback. We will incorporate the feedback in a newer version. Thank you.

**Withdrawal Confirmation:**

I have read and agree with the venue's withdrawal policy on behalf of myself and my co-authors.